# Understanding West Nile virus transmission: Mathematical modelling to quantify the most critical parameters to predict infection dynamics

Elisa Fesce[1]*, Giovanni Marini[2], Roberto Rosà[2,3], Davide Lelli[4], Monica Pierangela Cerioli[4], Mario Chiari[5], Marco Farioli[5], Nicola Ferrari[1,6]

1 Department of Veterinary Medicine and Animal Science (DiVAS), Wildlife Health management & One Health Lab, Università degli Studi di Milano, Lodi (LO), Italy, 2 Research and Innovation Centre, Fondazione Edmund Mach, San Michele all'Adige, Trento (TN), Italy, 3 Center Agriculture Food Environment, University of Trento, San Michele all'Adige, Trento (TN), Italy, 4 Istituto Zooprofilattico Sperimentale della Lombardia e dell'Emilia Romagna "Bruno Ubertini" (IZSLER), Brescia (BS), Italy, 5 Regional Veterinary Authority of Lombardy, Direzione Generale Welfare, Milano (MI), Italy, 6 Centro di Ricerca Coordinata Epidemiologia e Sorveglianza Molecolare delle Infezioni, Università degli Studi di Milano, Milano (MI), Italy

* elisa.fesce@unimi.it

**Data Availability Statement:** All relevant data are within the manuscript and its Supporting Information files.

## Abstract

West Nile disease is a vector-borne disease caused by West Nile virus (WNV), involving mosquitoes as vectors and birds as maintenance hosts. Humans and other mammals can be infected via mosquito bites, developing symptoms ranging from mild fever to severe neurological infection. Due to the worldwide spread of WNV, human infection risk is high in several countries. Nevertheless, there are still several knowledge gaps regarding WNV dynamics. Several aspects of transmission taking place between birds and mosquitoes, such as the length of the infectious period in birds or mosquito biting rates, are still not fully understood, and precise quantitative estimates are still lacking for the European species involved. This lack of knowledge affects the precision of parameter values when modelling the infection, consequently resulting in a potential impairment of the reliability of model simulations and predictions and in a lack of the overall understanding of WNV spread. Further investigations are thus needed to better understand these aspects, but field studies, especially those involving several wild species, such as in the case of WNV, can be challenging. Thus, it becomes crucial to identify which transmission processes most influence the dynamics of WNV. In the present work, we propose a sensitivity analysis to investigate which of the selected epidemiological parameters of WNV have the largest impact on the spread of the infection. Based on a mathematical model simulating WNV spread into the Lombardy region (northern Italy), the basic reproduction number of the infection was estimated and used to quantify infection spread into mosquitoes and birds. Then, we quantified how variations in four epidemiological parameters representing the duration of the infectious period in birds, the mosquito biting rate on birds, and the competence and susceptibility to infection of different bird species might affect WNV transmission. Our study highlights that knowledge gaps in WNV epidemiology affect the precision in several parameters. Although

**Funding:** This study was partially funded by the project MUSA - Multilayered Urban Sustainability Action (CUP H43C22000550001 to NF) and EU Horizon 2020 Framework Program, (grant 874850 MOOD to GM) and is catalogued as MOOD 22. The funders had no role in study design, data collection and analysis, decision to publish, or preparation of the manuscript. The contents of this publication are the sole responsibility of the authors and do not necessarily reflect the views of the European Commission.

**Competing interests:** The authors have declared that no competing interests exist.

all investigated parameters affected the spread of WNV and the modelling precision, the duration of the infectious period in birds and mosquito biting rate are the most impactful, pointing out the need of focusing future studies on a better estimate of these parameters at first. In addition, our study suggests that a WNV outbreak is very likely to occur in all areas with suitable temperatures, highlighting the wide area where WNV represents a serious risk for public health.

## Author summary

Infectious communicable diseases are currently one of the main burdens for human beings and public health. The comprehension of their spread and maintenance is one of the main goals to facilitate their control and eradication, but due to the complexity of their cycles and transmission processes, obtaining this information is often difficult and demanding. The control of vector-borne diseases in particular represents an important and very complex challenge for public health. Mathematical models are suitable tools to investigate disease dynamics and their transmission mechanisms and processes. To build a suitable model that can simulate transmission dynamics, a reliable and precise estimate of parameters for measuring transmission mechanisms is fundamental. We thus propose a sensitivity analysis of four unknown epidemiological parameters (bird recovery rate, mosquito biting rate, avian susceptibility to infection and avian competence to infection) that play a crucial role in driving West Nile virus (WNV) infection to determine which of them have the greatest impact on infection spread. This analysis suggests that the infectious period in birds and mosquito biting rate are the parameters to be prioritised in investigation to increase our ability to model WNV spread.

## Introduction

West Nile disease (WND) is an emergent vector-borne disease caused by West Nile virus (WNV), a single-stranded virus belonging to the Flavivirus genus [1]. Its cycle involves mosquitoes, mostly of the genus *Culex*, as vectors and diverse bird species as maintenance vertebrate hosts [2–5]. Although acting as dead-end hosts, several mammalian species, including humans, can be infected via mosquito bites and can develop symptoms ranging from mild fever to severe neurological disease [6,7]. Despite the low frequency of development of a severe illness (approximately 25% of infected persons develop symptoms [6]), recovery might take several weeks or months, and some effects on the central nervous system might be permanent [5,8]. In addition, recent diffusion of WNV in several countries in Europe and North America makes it one of the most widespread flaviviruses in the world [9]. In the Italian scenario in particular, WNV was detected for the first time in horses in the Tuscany region (central Italy) in 1998 [10], but is currently considered endemic in several Italian regions, making it a relevant public health issue. As a consequence, since 2013, health authorities have implemented an integrated surveillance system for early detection of its circulation in several Italian regions. The increased number of cases registered in 2018 increased attention to the need to develop an adequate surveillance system to monitor WNV circulation, to comprehend mechanisms and processes driving the spread of the infection and to fully understand the spatiotemporal variability. Despite that, the variety of bird species developing viraemia titres sufficient to infect feeding mosquitoes [11–13] and the differences in susceptibility and competence shown

between and within families of birds [14–17] make WNV spread and diffusion still poorly understood, especially on the European continent [18]. Few studies have provided a complete analysis of different bird species' viraemic responses, and the variety of the composition of local avian communities combined with the circulation of different WNV strains makes the extension of results to areas other than those tested potentially inaccurate [14,16,17,19,20]. To investigate spreading mechanisms and processes of WNV, several mathematical modelling efforts have been made (e.g., [21–26]), but the abovementioned uncertainties might hinder the precision of parameter estimates, thus impairing the reliability of simulations obtained and the likelihood of simulated mechanisms. For this reason, the development of specific studies to investigate the epidemiological effects of different parameters is fundamental to improve the reliability of model simulations and our comprehension of WNV spread, especially in Europe, where information about species involved in WNV spread and their epidemiological characteristics are currently lacking [18,19,27]. When a disease involves several species, particularly wildlife species, as in the case of WNV, field studies can be difficult and demanding. Each parameter may affect WNV dynamics in a different way; in fact, small changes in some parameters can produce a tremendous variation in transmission dynamics, while larger changes in other parameters can scarcely affect it. For this reason, a precise quantification of those parameters having the largest effect on disease dynamics can aid the prioritization of future research. Not least, Italy is characterized by a high variability of landscapes and climatic conditions, thus requiring the development of context-specific models to gain reliable clues on WNV transmission processes. In Lombardy region (northern Italy), surveillance against WNV is ongoing since 2013. Despite that, WNV is now endemic, thus implying the need to boost surveillance and detection techniques, but also to carry out new experiments to better grasp local circulation processes.

Our work provides a quantitative analysis of the relative contribution of specific transmission processes affecting WNV spread in Lombardy. We investigated the effects on pathogen transmission of different estimates of four epidemiological key parameters related to bird-mosquito-virus interactions, whose values are relatively unknown. We pursued this aim by deploying a previously published computational framework [26] informed with entomological, climatic, and ornithological data gathered in the Lombardy region between 2016 and 2018, and the investigated parameters were i) fraction of mosquito bites directed on competent birds, ii) avian competence, iii) avian recovery and iv) avian susceptibility to infection. We chose to focus on these four parameters because they are all related to birds, and the full understanding of WNV spread is hampered by the complexity between and within avian communities. Due to the variety of bird species and their inter-specific differences, WNV spread processes are very hard to investigate, both through experiments and modelling. Experimentation on birds (as vertebrates) is more demanding and usually relies on lower numbers if compared to the one on mosquitoes, furthermore laws protecting animal welfare make experiments on birds a limiting factor. For these reasons, disentangling the effect of bird-related factors on WNV transmission can be helpful both to understand the role of different bird species (and/or the avian community) and to address resources for further investigations.

## Materials and methods

### Dataset and reference system

Entomological data were collected between 2016 and 2018 in the Lombardy region in northern Italy (see Fig 1). Mosquito abundance records and their WNV infection prevalence estimated by the Regional WNV mosquito surveillance program performed by Regione Lombardia and Istituto Zooprofilattico Sperimentale della Lombardia e dell'Emilia Romagna (IZSLER).

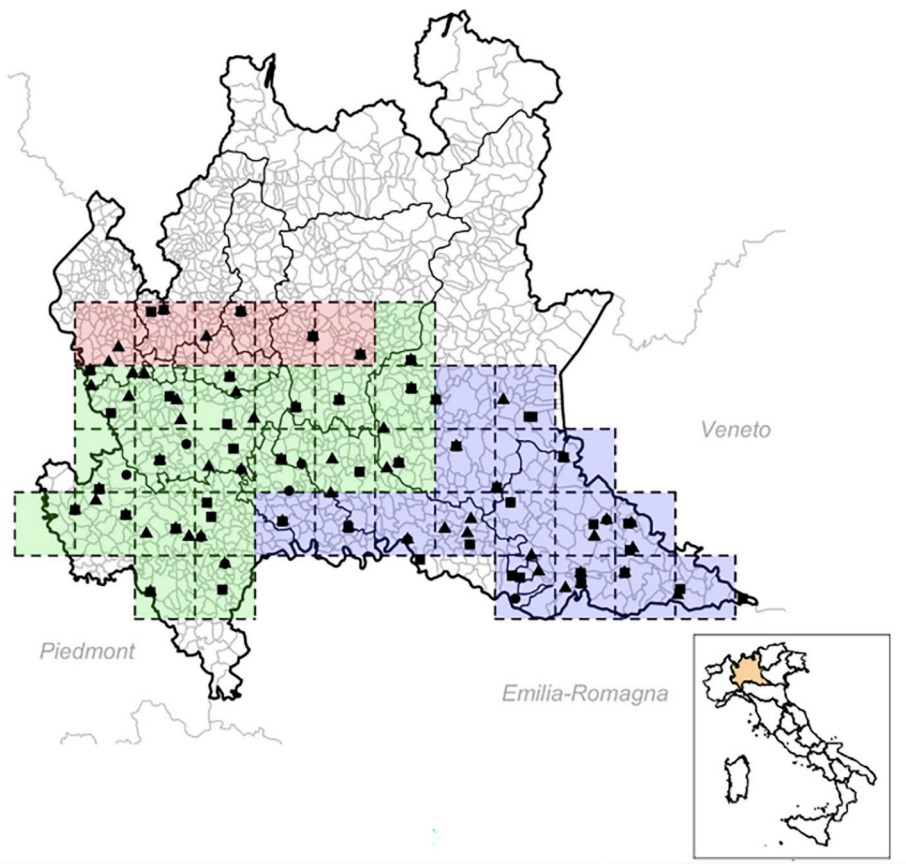

**Fig 1. Capture site division proposed for the Lombardy region.** Red quadrats represent the northern subregion, green quadrats represent the western subregion, and blue quadrats represent the eastern subregion. Black squares, circles and triangles represent mosquito capture sites for 2016, 2017 and 2018, respectively. This map was generated using R 4.1.2 (*raster* package, *http://gadm.org* source).

Mosquito abundance and positivity for WNV were determined by field collection and PCR analyses [28]. According to national and regional guidelines for the entomological surveillance of WNV [28–31], field collection of mosquitoes was carried out every two weeks between May and September by the use of $CO_2$ traps for capturing mosquitoes, covering an area of at most 400 km$^2$ each (Fig 1). Daily mean temperature data for each quadrat, collected with ground stations, were obtained from ARPA Lombardia (Agenzia Regionale per la Protezione dell'Ambiente della Lombardia). We then divided the Lombardy region into 3 separate subregions, a northern subregion, a western subregion and an eastern subregion (red, green and blue coloured regions in Fig 1), accounting for differences in temperature, precipitation, geography, mosquito abundance and WNV circulation. Due to the absence of WND in the northern subregion, we investigated WNV transmission only through the eastern and western subregions. To estimate the number of birds composing the whole avian community at the beginning of the summer and, in particular, the number of magpies, known to be a susceptible and competent avian species [19,32], we considered the records of the avifauna census provided by Regione Lombardia [33]. Specifically, the number of magpies was estimated as a proportion of the total number of birds estimated by a kriging method calculating the number of individuals circulating in an area $A = \pi^* r^2$, where r is the average *Culex pipiens* flight range (500 m, [24]).

## Model structure

The modelling framework followed that proposed in Marini et al. [26] to investigate WNV dynamics in the Emilia Romagna region. The model was adapted by fitting unknown model parameters on data collected in Lombardy region, thus allowing transmission quantification in this specific Italian region. Furthermore, given the importance of knowledge gaps about the contribution of different avian species to WNV spread, a parameter representing avian competence, was included among unknow epidemiological parameters. First, the mosquito population dynamics were simulated through an "*entomological model*", which provides a daily estimate of adult female mosquito abundance for each subregion and year in an average area $A$. Then, the estimated mosquito abundance was included in an "*epidemiological model*", aimed at simulating the transmission dynamics of WNV in one competent bird species. Due to their abundance and competence for WNV, as proposed by Marini et al. [26], mosquitoes of *Cx. pipiens* species were considered the only vector species. Analogously, as magpies are competent for WNV [19,28] and abundant in the Lombardy region, the avian host population was considered to grow and die with rates estimated for magpies [24]. The dynamics of the disease were simulated from May to October during 2016, 2017 and 2018 according to the data provided by entomological surveillance. For both the *entomological* and *epidemiological models*, the posterior distributions of the unknown parameters were explored following a Markov chain Monte Carlo (MCMC) method following the approach adopted in Marini et al. [26]. For both models, temperature-dependent parameters were estimated using the daily mean temperature for each quadrat, which originated from ARPA Lombardia. The full model framework is shown in Fig 2.

## The entomological model

To estimate *Cx. pipiens* abundance during the summer season, we calibrated the temperature-dependent entomological model presented in [26] on the recorded captures in the Lombardy region, averaged over each subregion. The posterior distributions of the unknown parameters (see below) were explored through an MCMC approach applied to the likelihood of observing the weekly number of trapped adults. Mosquito abundance was estimated for three years (2016, 2017 and 2018), starting from April up to October. The resulting mosquito abundance was then included as a known function $\omega(t)$ in the epidemiological model.

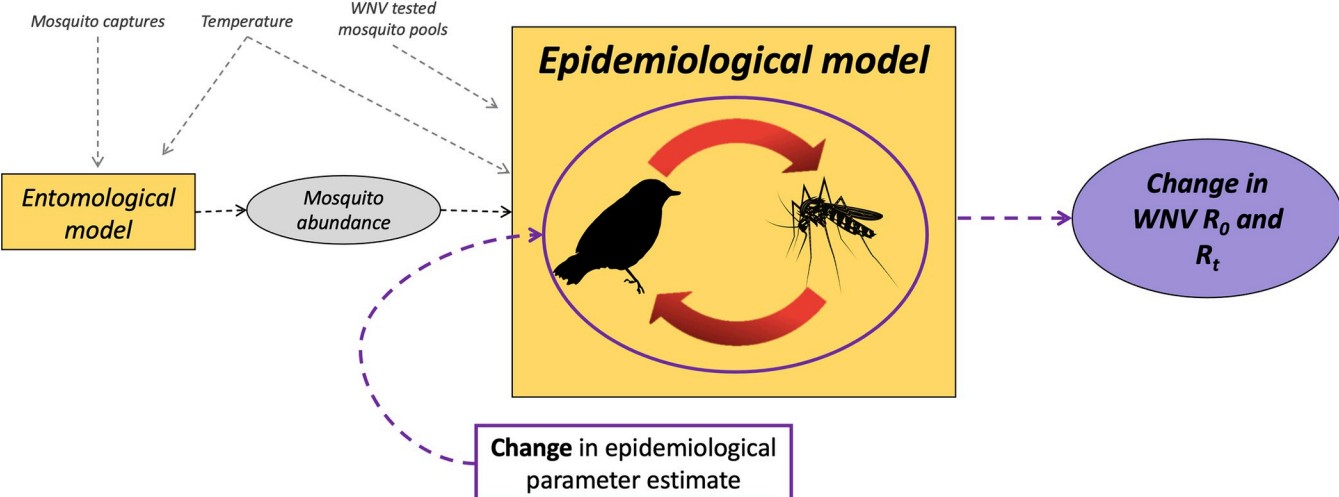

**Fig 2. Schematic representation of the computational framework.** Yellow squares: entomological and epidemiological models; grey circles: base model output; purple circle: sensitivity analysis output.

## The epidemiological model

The WNV epidemiological model is based on a system of eleven coupled differential equations, representing *Cx. pipiens* and a competent avian species divided into two age groups (juveniles and adults), and four infectious stages (susceptible, exposed, infectious and recovered). To account for the complexity of the avian population, the number of competent birds was estimated as a fraction ($a_i$) of the total number of birds estimated to live in the area ($B_0$). Birds (adults and juveniles, respectively) are considered to become infectious ($B_{Ia}$ and $B_{Ij}$) after an incubation period and then, as a consequence of infection, recover and become immune to reinfection ($B_{Ra}$ and $B_{Rj}$). At the start of the season (May), the avian population is considered to be fully composed of adult birds, reproducing and giving birth to juvenile birds until mid-July. Because the maturity age of birds can be considered one year [34], the newly born birds are considered juveniles throughout the entire season. Mortality due to infection in birds was neglected because of the limited mortality due to WNV infection observed in magpies in Europe [2]. For the sake of simplicity other transmission routes as bird-to-bird and vertical transmission within birds, were not included in the model. Mosquitoes are considered to become infectious after a temperature-dependent incubation period ($\theta_M$), and they remain infectious for the rest of their life. Birds can acquire infection according to their susceptibility ($p_{MB}$) and to a mosquito biting rate over the competent bird population ($b$). The biting rate was defined as follows [35]:

$$b = f \cdot 0.122 \cdot log(T - 9)^{1.76}$$

Were $T$ is the mean daily temperature and $f$ is the fraction of bites of mosquitoes directed to the competent bird species. In the epidemiological model we considered two different fraction of bites ($f$): $f_1$ for the initial part of the season and $f_2$ for the latest part of the season because of the observed shift in the fraction of bites on competent birds between the early and late seasons [36]. The infection rate of mosquitoes was estimated by a temperature-dependent function representing the probability of WNV transmission from infectious birds to mosquitoes per bite ($p_{BM}$, [24]). Viremia titres developed after infection can vary widely among bird species [17]. Avian competence is considered the probability of a mosquito becoming infectious after biting an infectious bird; thus, it was modelled as scalar in the interval 0–1 ($p$) multiplying the probability of WNV transmission from birds to mosquitoes ($p_{BM}$). The avian recovery rate ($v_B$) is represented by the inverse of the duration of the infectious period in birds.

The complete scheme of the model and the equations describing the system are reported in the epidemiological model structure paragraph within the S1 Appendix.

## Free parameter estimate

All biological and epidemiological parameters used for birds and mosquitoes related to magpies and *Cx. pipiens* are those reported in Marini et al. [26]. The posterior distribution of unknow population and epidemiological parameters ($\Psi$) were estimated through a Monte Carlo Markov Chain sampling, applied to the likelihood of observing the recorded number of mosquitoes in the entomological model (Poisson distribution) and the number of positive pools in the epidemiological model (binomial distribution). More specifically, the likelihood function for the epidemiological model follows the one proposed in [26]:

$$L = \prod_{y \in \{2016, 2018\}} \prod_{s=1}^{N_s(y)} \binom{N(s,y)}{K(s,y)} P(s, y, \Psi)^{K(s,y)}$$

Where $N(s,y)$ and $K(s,y)$ are respectively the number of tested and positive pools in year $y$ at sampling time $s$. The probability P was estimated as follows:

$$P(s, y, \Psi) = 1 - \left( 1 - \frac{M_I(s, \Psi)}{M(s, \Psi)} \right)^{u(s)}$$

Where $M_I(s, \Psi)$ and $M(s, \Psi)$ are the number of infected and total mosquitoes predicted by the model for the sampling date s and model parameters $\Psi$. u(s) represents the mean pool size.

For both models, a total of 100,000 iterations of the MCMC sampling were performed, and the posterior distribution of $\Psi$ was obtained by using random-walk Metropolis-Hastings sampling approach and normal jump distributions. A burnin of 10% and thinning (N = 100) was applied to reduce autocorrelation between parameter estimates; the resulting posterior distribution and the analysis of autocorrelation of parameters are reported in Supplementary Materials (S1 Appendix). Convergence of the MCMC was checked by visual inspection of posterior distribution of parameters.

The unknown parameters estimated by the MCMC method for the entomological model are:

- $K_1$: density-dependent scaling factor driving the carrying capacity for the larval population at the beginning of the season

- $K_2$: density-dependent scaling factor driving the carrying capacity for the larval population at the end of the season

- $M_0$: the number of mosquitoes at the beginning of the season

We considered two different scaling factors for carrying capacities, as during summer *Cx. pipiens* breeding site availability might change, causing a possible increase in larval mortality, for instance, because of competition for resources with *Ae. albopictus* at the larval stage [37].

For the epidemiological model, the unknown parameters are:

- $a_i$: the proportion of competent birds out of the total avian population in subregion $i$

- $f_1$: the fraction of mosquito bites on the competent avian population at the beginning of the season (from May to mid-July).

- $f_2$: the fraction of mosquito bites on the competent avian population at the end of the season (from mid-July to October).

- $i_b$: the number of immune-competent birds in each subregion at the beginning of the first year of simulations (2016)

- $p$: avian competence, defined as the probability for a competent infectious bird to transmit the infection to a mosquito

- $p_{MB}$: bird susceptibility to infection, considered as the probability for a competent bird to become infected when bitten by an infectious mosquito

- $v_B$: the bird recovery rate, considered as the inverse of the duration of viraemia

All prior distributions of epidemiological parameters are reported in Table 1.

As a characteristic of the infection itself, all epidemiological parameters were considered constant in time and space and were estimated across all years and subregions. Only the number of immune-competent birds was considered different among years and between subregions. The number of immune-competent birds was estimated for 2016 (the first year of simulation), and for the following years, it was considered dependent on the level of avian

**Table 1. Estimated model parameter distributions (average and 95% quantiles).**

| Parameter | Parameter biological meaning | Prior distribution | Estimate range (2.5%-97.5% percentile) |
|---|---|---|---|
| $B_0$ | Initial number of birds (whole avian community) | $U(50,70)$ | 60.20 (50.44–69.57) |
| $a_i$ | Proportion of competent birds among the whole avian community | $U(0,1)$ | 0.25 (0.003–0.91) [1] |
| $i_{Bw}$ | Proportion of immune birds at the beginning of the first simulated season, western subregion | $U(0,1)$ | 0.54 (0.03–0.99)[1] |
| $i_{Be}$ | Proportion of immune birds at the beginning of the first simulated season, eastern subregion | $U(0,1)$ | 0.54 (0.04–0.97)[1] |
| $f_1$ | Proportion of mosquitoes' bites directed to the competent avian species during the early season | $U(0,0.5)$ | 0.35 (0.091–0.49)[1] |
| $f_2$ | Proportion of mosquitoes' bites directed to the competent avian species during the late season | $U(0,f_1)$ | 0.28 (0.025–0.49)[1] |
| $p$ | Competence of the competent avian population | $U(0,1)$ | 0.65 (0.14–0.98)[1] |
| $p_{MB}$ | Susceptibility of the competent avian population | $U(0,1)$ | 0.69 (0.19–0.99)[1] |
| $v_B$ | Recovery rate | $U(0,1)$ | 0.22 (0.002–0.94)[1] |

[1] MCMC estimate

immunity estimated at the end of the previous year. The total initial number of birds was randomly chosen in the range of 50–70 to account for the variability in avian abundance across sites and years. Mosquito WNV prevalence at the beginning of each season was randomly chosen in the range of 0–0.001 [26].

Finally, to verify whether there was a statistical relationship among unknown epidemiological parameters, we checked for their mutual correlation (Pearson correlation).

## Basic reproduction number ($R_0$)

We estimated the WNV basic reproduction number ($R_0$) following the formula proposed in [38] for vector-borne diseases. In this system, $R_0$ represents the average number of secondary infected mosquitoes over the entire transmission cycle, following the introduction of an infected mosquito into fully susceptible mosquito and bird populations. The formula takes into consideration both the number of secondary infected birds following the introduction of an infectious mosquito into a fully susceptible bird population and the number of secondary infected mosquitoes following the introduction of an infectious bird in a fully susceptible mosquito population.

The basic reproduction number then computed as:

$$R_0 = R_0^{MB} \cdot R_0^{BM}$$

with

$$R_0^{MB} = \frac{b \cdot p_{MB}}{\mu_M}$$

$$R_0^{BM} = \frac{b \cdot p \cdot p_{BM}}{v_B} \cdot \frac{\theta_M}{\mu_M + \theta_M} \cdot \frac{M_S}{N_B}$$

$R_0^{MB}$ ($R_0^{BM}$) is the reproduction number of WNV and represents the number of hosts (mosquitoes) infected by an infectious mosquito (host). Following the formula proposed in [38], the $R_0$ estimate depends on the transmission probability (here $b \cdot p_{MB}$), the adult mosquito infectious lifespan ($\frac{1}{\mu_M}$), the initial number of mosquitoes per bird ($\frac{M_S}{N_B}$, where $M_S$ is the number of susceptible mosquiteos and $N_B$ the total number of competent birds), the probability of surviving latent period for a mosquito ($\frac{\theta_M}{\mu_M + \theta_M}$) and the duration of infectious period in birds ($v_B$).

By definition of $R_0$, here the number of susceptible mosquitoes $M_S$ equals the total number of mosquitoes. Considering the wide variability of mean daily temperatures between April and October in the study area (which can easily range from 14 to 35˚C, according to ARPA Lombardia records) and the great effect that temperature has on model simulations by affecting the mosquito death rate ($\mu_M$), the probability of WNV transmission from infectious birds to mosquitoes ($p_{BM}$) and the mosquito biting rate (b) [39–41], we arbitrarily chose to perform all simulations to estimate $R_0$ and $R_t$ by using a fixed temperature of 24˚C.

### $R_t$ estimate

To estimate the potential of WNV to spread during the summer season, we adapted the formula proposed for $R_0$ to estimate the effective reproduction number $R_t$, defined as the number of new infections caused by a single infected individual at time $t$ in a partially susceptible population.

The effective reproduction number is thus computed according to the following formula:

$$R_t = R_t^{MB} \cdot R_t^{BM} = R_0^{MB} \cdot R_0^{BM} \cdot \frac{B_S}{N_B}$$

where $R_t^{MB}$ ($R_t^{BM}$) is the number of hosts (mosquitoes) infected by an infectious mosquito (host), $B_S$ is the number of susceptible birds, and $N_B$ is the total number of competent birds. As a consequence, the $R_t$ estimate depends on the vector-host ratio ($\frac{M_s}{N_B}$) and on the proportion of susceptible hosts over the whole host population $\frac{B_S}{N_B}$.

### Transmission spread and maintenance during summer season

To investigate the transmission of WNV throughout the summer season, for each set of parameters estimated by the MCMC procedure, a daily $R_t$ was estimated from May to October, accounting for the daily mean temperature. Then, considering $R_t = 1$ as a necessary and sufficient condition for WNV to spread in the mosquito population, we calculated the monthly frequency for $R_t$ to be above 1 on the overall number of simulations performed. We will refer to the above mentioned frequency as outbreak frequency.

### Sensitivity analysis of unknown epidemiological parameters

Sensitivity analysis was performed on the following parameters:

- Fraction of mosquito bites on competent birds ($f$)

- Competence for WNV of the bird species ($p$)

- Competent birds' susceptibility ($p_{MB}$)

- Duration of competent birds' infectious period (recovery rate, $v_B$)

To investigate the effect of different parameter sets, we performed a sensitivity analysis by varying each epidemiological parameter estimate in turn and evaluating how that change affects $R_0$. Each changing parameter was allowed to vary into the 2.5%-97.5% percentile of its posterior distribution, whereas all other parameters were considered fixed to their mean value. To estimate $R_0$, the number of susceptible mosquitoes (and susceptible birds or total birds) was estimated as the mean number of susceptible mosquitoes (susceptible birds or total birds) in each subregion and year over the entire season.

For a deeper comprehension of the effect of a parameter variation, we also estimate how the outbreak frequency (i.e., the number of times the estimated $R_t$ is above one divided by the total

number of simulations performed every month) changes according to a change in parameter estimate. To estimate $R_t$, the number of susceptible mosquitoes (and susceptible birds or total birds) was estimated as the daily mean number of susceptible mosquitoes (susceptible birds or total birds).

### Temperature and host-vector ratio effect on $R_0$

Finally, to highlight the effect of temperature on WNV spread, we performed a sensitivity analysis by varying the mean daily temperature from 10 to 30˚C (with a step of 0.1˚C) and estimating $R_0$ for each temperature for each set of parameters used for simulations. Then, the WNV outbreak frequency at different temperatures was estimated as the frequency for $R_0$ to lie above 1. Although temperature affects mosquito abundance during the season, to obtain more generalizable results, we investigated the temperature effect considering only four different values of vector-host ratios (i.e., 100, 1000, 7256 and 10,000 mosquitoes/bird). We chose 7256 as it is the mean vector-host ratios over the entire season obtained from the simulations.

## Results

### Model calibration and fit

Estimates show a very high variability for all investigated parameters. (Table 1). The fractions of bites on competent birds (both $f_1$ and $f_2$) are the parameters showing the lowest variability, followed by the avian competence and susceptibility (p and $p_{MB}$ respectively). A full list of the parameter estimates (and their estimated range) obtained by the MCMC approach is reported in Table 1. The 95% quantiles of the model predictions include 97% of the observed points, showing that despite the large variation in model predictions, the model can well describe WNV dynamics in the Lombardy region. For further details about the model fit and obtained simulations, see Fig B in S1 Appendix.

No meaningful correlation was observed (absolute coefficient values lower than 0.5, P>0.05) among all unknown parameters, except between proportion of immune birds at the beginning of the first simulated season in the western and eastern subregions ($i_{Bw}$ and $i_{Be}$, corr = 0.98, P<0.001) and between recovery rate and the proportion of competent birds ($\nu_B$ and $a_i$, corr = -0.54, P<0.001) (Fig F in S1 Appendix).

### Transmission maintenance during seasons

The simulated seasonal pattern of the outbreak frequency of WNV in the mosquito population has an initial probability value of 0.47 in May followed by an increase to 0.99 in June and then a continuous decrease from July (0.97) to October (0.024) (Fig 3).

### Effect of unknown epidemiological parameters on $R_0$

All the investigated epidemiological parameters were found to affect the $R_0$ estimate, with a trend that is not affected by differences between subregions and years (represented with different colours in Fig 4). Of the four parameters that have been investigated, the highest range of change for $R_0$ was produced by a change in the duration of infectious period in birds ($\nu_B$), followed by fraction of bites on birds (f), then the birds' competence (p) and birds' susceptibility ($p_{MB}$) (Table 2). The effect on $R_0$ estimate was not linear for the fraction of bites on birds (f), that showed a higher effect for higher values of f, and for the duration of infectious period in birds ($\nu_B$), that had a higher effect for lower values of $\nu_B$ (Fig 4A and 4D). The avian competence (p) and susceptibility ($p_{MB}$) instead, showed the same linear increase (Fig 4B and 4C). The visual inspection of pairwise perturbation of parameters (Fig H in S1 Appendix) showed

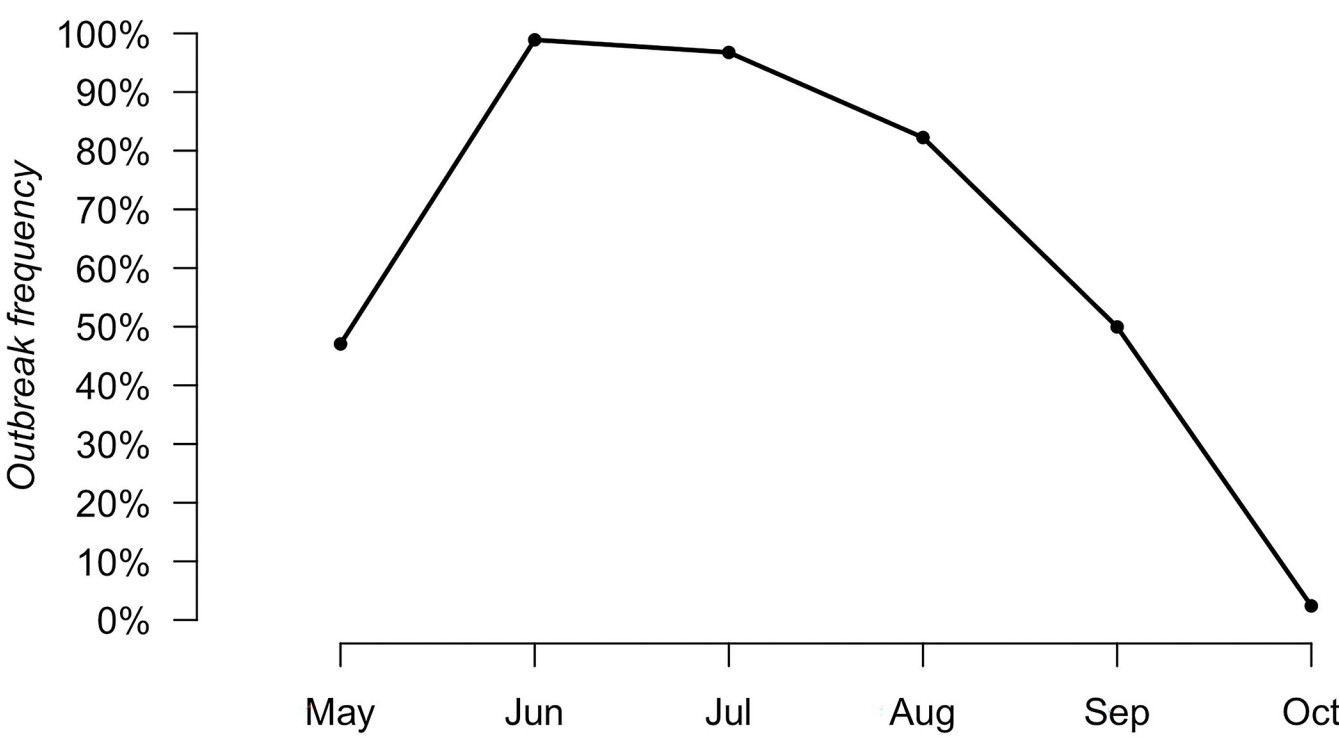

**Fig 3. Simulated outbreak frequency of WNV into the mosquito population during the summer season.**

no combined effect of parameter variation. The outbreak frequency is highly affected by a change in the fraction of bites on birds ($f$, Fig 5A) and to a lesser extent by a change in the duration of infection in birds ($v_B$, Fig 5D). Bird competence ($p$) and susceptibility ($p_{MB}$) have the lowest effect on the outbreak frequency (Fig 5B and 5C). For all parameters investigated a change in their estimate can result in a shift of the month with the highest outbreak frequency from June to July (Fig 5).

### Temperature and host-vector ratio effects on $R_t$

As expected, $R_t$ is affected by temperature. Different vector-host ratios ($\frac{M_S}{N_B}$) strongly impact the $R_t$ estimate, but show a similar trend of the effect of temperature on $R_t$. In accordance with previous analyses, for all tested vector-host ratios, $R_t$ was always equal to zero for any tested temperature below 14˚C. For $\frac{M_S}{N_B} = 10,000$, Rt rapidly increased reaching 100% of simulations with $R_t > 1$ at 16.1˚C (long dashed line). Decreasing the vector-host ratio lowered the increase in Rt with temperature, reaching 100% at 19.5˚C and for $\frac{M_S}{N_B} = 1,000$ (Fig 6, dashed line). $\frac{M_S}{N_B} = 100$ reaches an outbreak frequency of 96% at 26.1˚C (Fig 6, dotted line). With a vector-to-host ratio equal to 7256 (the mean vector host-ratio of all simulations) (Fig 6, red solid line), the outbreak frequency reaches 100% of simulations at a temperature of 16.4˚C.

### Discussion

With the present study, we explored the effect that the selected parameter estimates have on the estimate of WNV transmission. Due to the existence of several knowledge gaps regarding species and infection processes involved in WNV spread, further field and laboratory investigations would improve our possibility to successfully forecast the dynamics and spread of this

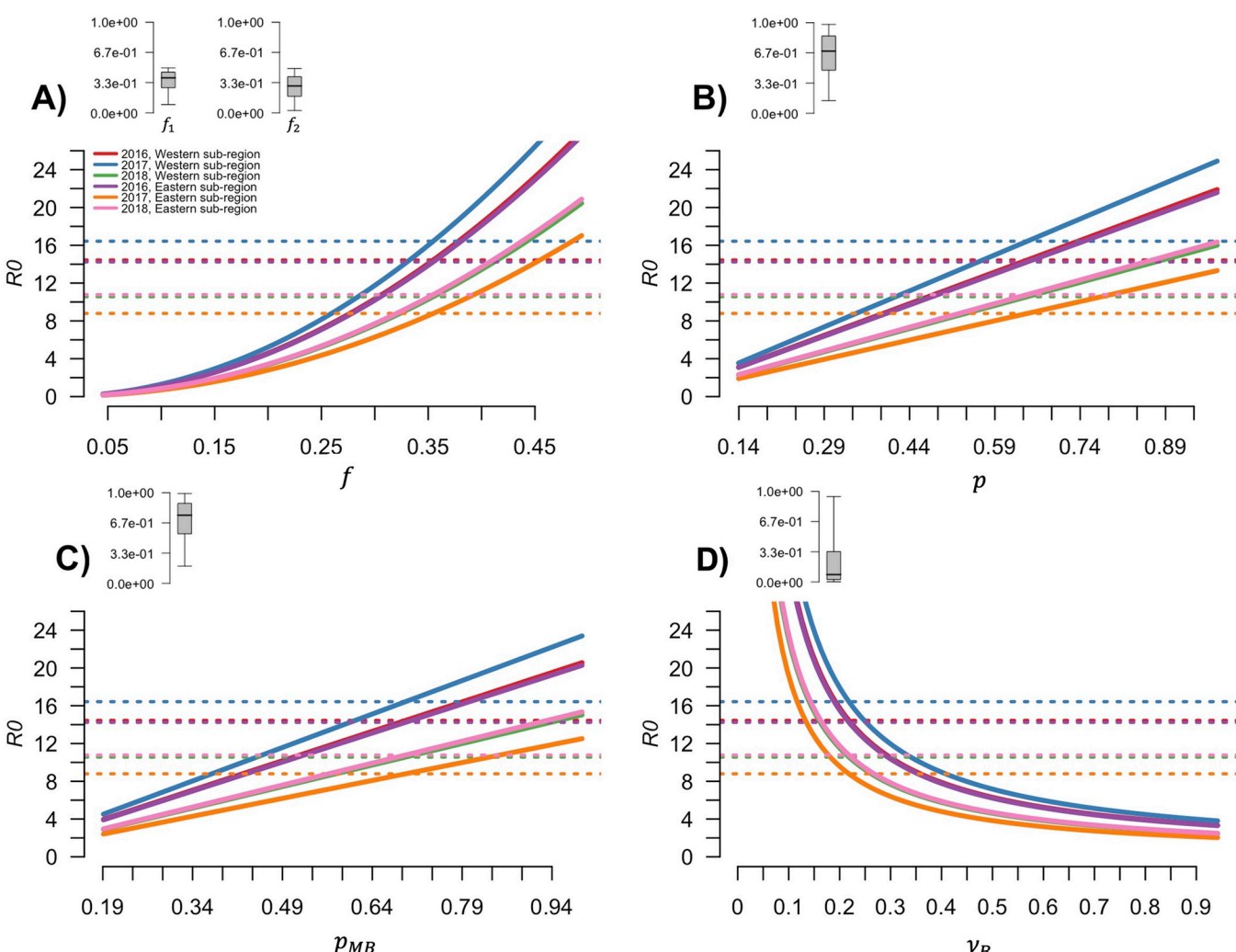

**Fig 4. Effect of different changes in model parameters on the $R_t$ ratio.** Effect of A) fraction of bites on competent birds ($f$); B) bird competence (p); C) bird susceptibility to infection ($p_{MB}$); and D) duration of infectious period in birds ($v_B$) on $R_0$ in each sub-region and year of simulation (solid lines). Dotted lines represent the mean value of $R_0$. The grey boxplot provided for each panel visualises the posterior distribution of the parameter investigated. The posterior distribution of parameters is visualised using the same y axis to compare the shown distributions.

pathogen [18]. As field and laboratory studies can be very expensive and time consuming, we investigated through mathematical modelling which of the four selected epidemiological parameters might have the largest effect on our predictions. By disentangling which of the investigated parameters affected more our model simulations, we aimed at providing a quantitative measure of the beneficial effect of a more precise estimate of each parameter tested. By

**Table 2. Ranges of $R_0$ estimates.** $R_0$ values were calculated at 2.5% and 97.5% percentile of parameter estimates for each considered parameter, while others were assumed equal to their estimated average.

| Parameter | $R_0$ range |
|---|---|
| $f$ | 0.21–24.29 |
| $p$ | 2.7–19.01 |
| $p_{MB}$ | 3.45–17.85 |
| $v_B$ | 2.91–1058.24 |

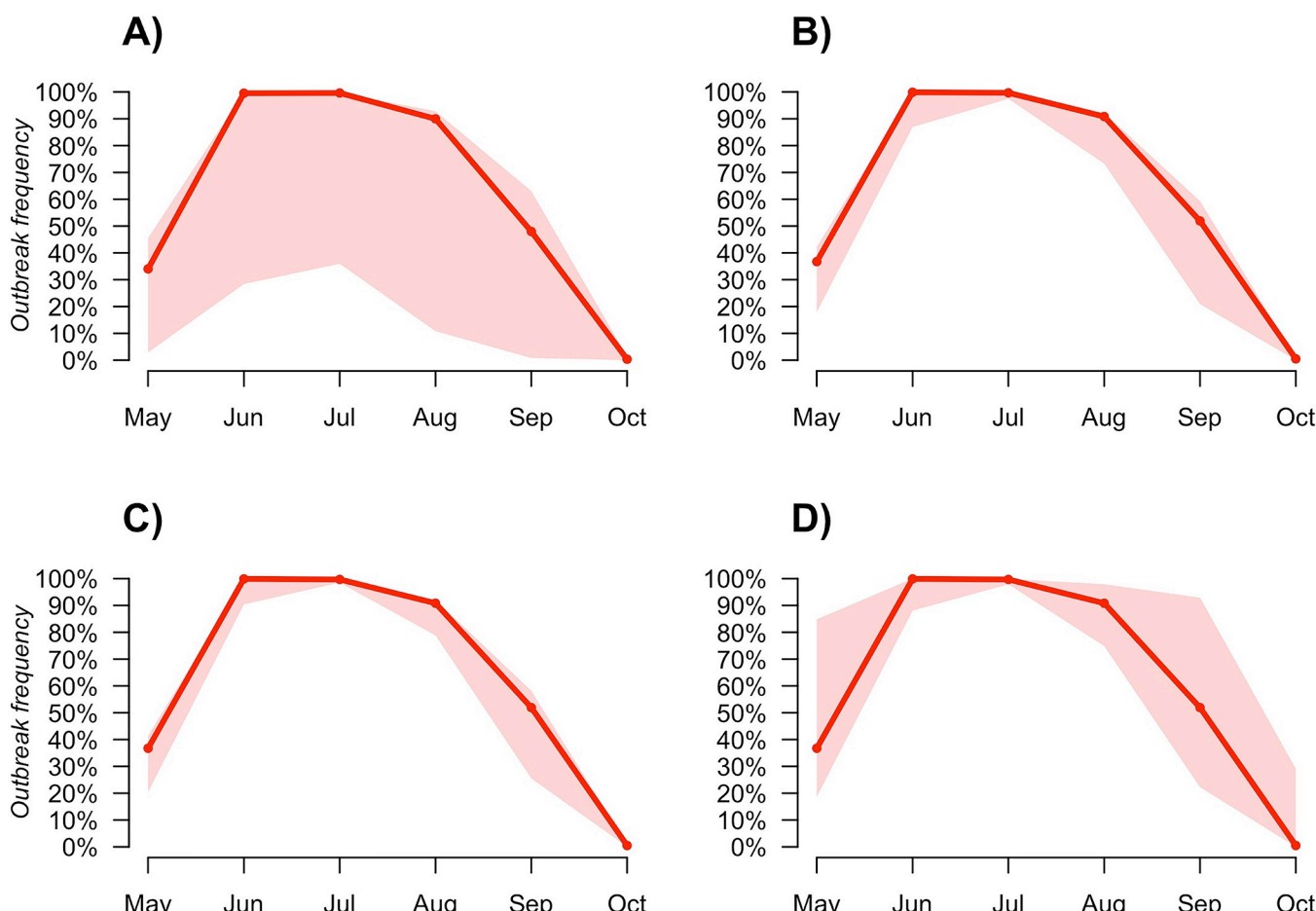

**Fig 5. WNV outbreak frequency in the mosquito population according to parameters variability.** WNV outbreak frequency in the mosquito population during the summer season depended on the variation in A) fraction of bites on competent birds ($f$); B) bird competence ($p$); C) bird susceptibility ($p_{MB}$) and D) duration of infectious period in birds ($v_B$) estimates. The solid red line represents the outbreak frequency estimated using the mean value of the parameters' posterior distributions. The red-shaded area shows the outbreak frequency obtained assuming all free parameters but one (allowed to range in the 95%CI) equal to their average.

calibrating our model with entomological and ornithological data gathered in northern Italy, we focused on a specific area, but results are easily generalizable to other European areas. From the sensitivity analysis performed, we observed that the duration of the infectious period in birds ($v_B$) is the parameter that has the largest effect on disease dynamics, and its effect is bigger for long durations of infection (i.e. small values of $v_B$). Considering that the viraemic period has been estimated to lasts between 2 and 5 days (i.e. $v_B$ in 0.2–0.5) according to one laboratory experiment [17], we highlighted the importance of having a reliable and precise estimate of the bird recovery rate. The second most impactful parameter is the fraction of bites on competent birds ($f$), that increases its influence when increasing its value. This result shows that a correct estimate of this parameter is essential to predict WNV spread. As the effect on $R_0$ increased with the increase of this parameter value, we showed that the more frequently the competent bird species is bitten, the more important is to precisely estimate this frequency. This parameter also seemed to be the most relevant when estimating the outbreak frequency during the season. Low estimates of $f$ drastically decrease the predicted outbreak frequency, thus possibly leading to an underestimate of the spread of the infection. These results then imply the importance of a correct estimate of *Cx. pipiens* feeding preferences through field

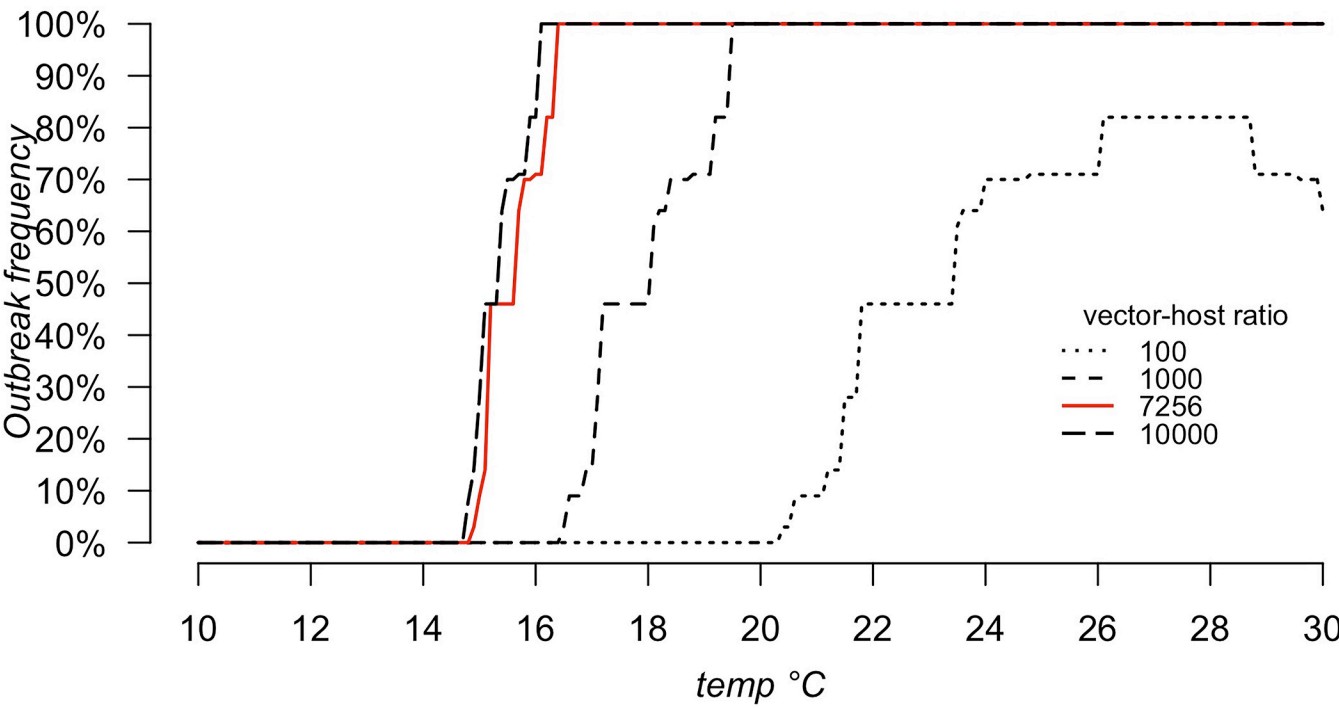

**Fig 6. Outbreak frequency as a function of temperature (10–30˚C) and different vector-host ratios.**

experiments (e.g. [36]), but also suggest that species that are more frequently bitten might have a key role for WNV spread.

The susceptibility to infection ($p_{MB}$) and the competence ($p$) of birds instead, showed a lower and linear effect. Despite the similar results obtained for these two parameters, we analysed them separately because of their very different biological meaning.

In addition, according to previous findings [42], temperature highly affects outbreak frequency, not allowing WNV to spread when the daily mean temperature is lower than 14˚C. The Lombardy region, our study area, shows all suitable characteristics to allow WNV spread during summer, especially in June, when the environmental conditions seem to ensure the possibility of WNV spread in the mosquito population.

Despite the progress in health care and preventive measures, interventions to control the spread of infectious communicable diseases remain one of the main goals for public health [43,44], but this goal is often impaired by the lack of information and certainties about transmission processes driving infection transmission dynamics [18,45]. Mathematical modelling can be an efficient tool to investigate infection dynamics and transmission processes and mechanisms (e.g., [46] and [47]), but the adoption of this approach requires robust parameter estimates to be reliable, and it has often been hampered by existing limitations due to inadequate or partial data availability. If, on the one hand, the development of field and laboratory investigations is fundamental to enhance our comprehension of transmission processes, on the other hand, this process can be very long and demanding, making a prioritization of the most useful investigations essential. Nonetheless, transmission processes do not have the same impact on disease dynamics. Indeed, a small change in some parameter estimates can cause a large variation in model predictions, whereas large variations in other parameters can result in a smaller variation in model prediction [48]. Therefore, before designing any field study, it can

be helpful to identify those parameters that have the largest impact on the infection dynamics and hence require primary attention.

WNV is now considered one of the most widespread arboviruses in the world, with human cases identified worldwide [49]. Nevertheless, there are several processes and mechanisms that drive its spread and maintenance into wild populations, which are still unknown [18]. Many species are considered suitable as hosts or vectors, thus possibly giving different contributions to infection dynamics depending on individual and species-specific characteristics [17,19,23,50]. Moreover, the variety of involved species, the different patterns in virus transmission cycles, and the existing differences in avian community composition among different areas [13,20,51] further contribute to increasing the sources of variability in infection dynamics, thus making it difficult and expensive to collect field data required to fill knowledge gaps. Indeed, antibodies against WNV have been detected in a broad range of wild and domestic bird species worldwide, and the virus has been isolated from different avian species [13]. In addition, viraemic titres developed by different birds have been shown to strongly depend on both the host species and virus lineage [16,17,19,20]. Furthermore, mosquito feeding behaviour can change among areas and mosquito species, depending on both host abundance and mosquito feeding preference [36,52,53]. All these characteristics are suitable to drive or influence WNV spread.

In this context, the present work aimed to reveal how a change in a set of WNV epidemiological parameters can affect our estimate and understanding of infection spread. We found that changes in the recovery rate and in the fraction of bites directed on competent birds affect Rt estimate more profoundly than a change in the susceptibility of the avian species or avian competence. We highlighted the importance of having a precise estimate for the bird recovery rate, showing that a change in this parameter value, especially if we consider low recovery rates (i.e., long infectious period duration), widely affects the basic reproduction number of the infection $R_0$ (i.e., the number of secondary infected mosquitoes in fully susceptible mosquito population). On the one hand, these results highlight the need for a careful estimate of species-specific recovery rates to predict WNV spread and dynamics. On the other hand, they point out the need to compare species-specific rates to determine which of the investigated avian species plays the major role in spreading the infection. According to model simulations, different estimates of the fraction of mosquito bites on competent birds can also widely affect the basic reproduction number ($R_0$) of WNV, also having an effect on the outbreak frequency during summer. The interaction between birds and mosquitoes is known to play a central role in disease spread [54]; moreover, it has been shown that mosquitoes can selectively choose where to feed, preferring specific species to others [36,55]. According to our results, since the effect of the fraction of bites on competent birds estimates is important, and particularly high for increasing fraction of bites, we can conclude that understanding which conditions and species-specific characteristics drive the probability of being bitten is critical to fully understand and predict the spread of WNV. In addition, this result again helps us to identify avian species that play an important role in WNV transmission, such as those highly bitten or preferred by mosquitoes. Furthermore, by pointing out the role of the fraction of bites on competent birds in infection spread, this result highlights the importance of investigating the extent to which mosquitoes actively choose which species to feed on and the extent to which the fraction of bites is driven by species abundance. For example, in northern Italy, blackbirds are frequently bitten by mosquitoes, even if they are less abundant than other species [36]. This finding, coupled with the high number of blackbird individuals living in the area, could suggest their role in the spread of WNV. Being aware of which avian species are primarily involved in the spread of infection would help us to fill some of the current knowledge gaps and improve our understanding of WND and would allow us to efficiently estimate and predict the risk of infection for human beings and consequently to develop appropriate intervention strategies to reduce it.

Avian competence, despite the high species-specific differences reported [17], seems to have a lower impact on the spread of the infection. It is assumed that to be capable of infecting mosquitoes (*Cx. pipiens*), birds need to develop viraemic titres greater than $10^5$ PFU/mL [5], but the collection of this information can be logistically demanding and hard to perform for wild birds. Moreover, experimental infections might not successfully mimic the natural infection occurring in wildlife, as mosquito inoculations of WNV in birds can result in higher viremias than needle injections [56]. Consequently, the efforts required to estimate bird competence may be greater than the benefit obtained.

As anti-WNV antibodies have been detected in several bird species [16,17,32,57], a wide variety of birds may be considered susceptible to WNV infection. Bird surveillance, ongoing in several countries, can therefore be useful to help identify susceptible and potentially competent bird species. Despite that, our analysis showed that avian susceptibility to WNV infection has a small effect on both the number of secondary infected mosquitoes and on the outbreak frequency of the disease. This finding implies that, despite being informative on the circulation of WNV in the avian population, the investigation of the WNV positivity of birds cannot be considered one of the most significant studies to be performed. Furthermore, this work supports the hypothesis that temperature and competent mosquito presence are limiting factors for WNV spread [39–42,58–60]. Indeed, according to model simulations, the $R_t$ of the infection changes following a change in temperature and in the vector-host ratio. Despite that, the effect of all epidemiological parameter estimates does not change for changing temperature and vector-host ratio, making our findings generalizable and extendible to other areas with similar environmental conditions. It is also important to note that in our study area, the mean daily temperature between May and September, and the recorded mosquito densities in those months can sustain WNV spread, highlighting the high human infection risk in this area. In the present work we considered the basic reproduction number of WNV ($R_0$) as the number of secondarily infected mosquitoes, after the introduction of one infectious mosquito in the system. It follows that we are accounting for the number of birds that can get infected after the introduction of an infectious mosquito, and then on the number of mosquitoes that are infected at the end of one cycle of the infection. As a consequence, $R_0$ will depend on both the number of susceptible birds and susceptible mosquitoes at the begin of the cycle (included in the $R_0$ formula as a vector-host ratio $\frac{V}{H}$, that gives us the number of vectors per host we expect). According to our simulations, for some set of parameters, 100 mosquitoes/host can be enough for $R_0$ to be larger than zero, allowing WNV to spread. We considered a deterministic model, which might possibly overestimate WNV circulation at the beginning of the season, since with few infected individuals it could be a combination of largely stochastic events. Nonetheless, this result suggests that regardless the values of epidemiological parameters, suitable conditions of temperature and of mosquito abundance might sustain WNV transmission in the area. It follows that we should not neglect the possibility of WNV to spread in new areas with suitable conditions, especially considering the possible long flying distance of birds, raising the question if a surveillance plan could also be beneficial in nonendemic areas with suitable climatic conditions. Moreover, we can observe that the proportion of competent birds necessary to maintain the infection ($a_i$) estimated from posterior distributions of parameters is very variable, ranging from few birds on a population of 50–70 individuals to the 90% of the population. Having considered one species only, from this model alone we cannot provide definitive conclusions about the prevalence of WNV in the whole avian community. Despite that, this result raises the question if an early WNV circulation can be detected in birds. On the one hand indeed if only few birds are competent and infectious, we might miss them during the surveillance. On the other hand, as information on the competence for WNV transmission of

European species is scarce, we might overestimate WNV circulation by testing not competent species. On the contrary, despite the low WNV prevalence estimated in mosquitoes, we believe that that entomological surveillance, allowing faster and easier collection of numerous samples, is likely to be more informative for an early detection of WNV. Despite that, the investigation of the most efficient surveillance plan to detect WNV spread is beyond the scope of this work and further investigations are needed to evaluate it.

It is necessary to note that one limitation of the present model is the assumption of having only one competent avian species. Despite this oversimplification, which could be overcome by future studies, this modelling approach has a proven ability to simulate and investigate WNV spread in nearby regions (i.e., Veneto and Emilia Romagna), suggesting its reliability despite its limitations [24,26].

We need to acknowledge that the range of knowledge gaps currently hampering our comprehension of WNV spread is not limited to the investigated four parameters. Several other aspects, like the role of humidity, the effects of diurnal temperature variation or the duration of immunity in birds might sensibly affect WNV spread, and our model only tries to cover a small part of a wider scenario of unknowns and gaps. Despite there are still several gaps that needs to be filled, disentangling which of these four parameters needs to be further investigated is crucial in the European context. Studies performed on American bird species, showed a high inter-species variability in most of epidemiological characteristics [17]. WNV has been found in several bird species (birds of prey included), showing its ubiquity in the avian community. Furthermore, due to differences between species and viral strains circulating in the two continents, most of the estimates provided for American birds cannotft be simply transferred to European species. Thus, given the variety of bird species living in European countries, their differences with American species, and the fragmented and partial information available for European species, our results provide a criterion to assess future investigations on WNV.

Despite that, we acknowledge that our results depend on model choices, and further investigations through different models might provide additional insights into WNV dynamics. Among the limitations of this model, it is necessary to notice how after fitting the model on data, the posterior distributions of parameters still showed a very high variability, suggesting the need of further explorations such as a restriction the space of existence of some of the parameters with values estimated from literature. As few studies have been performed in Europe in this regard, and species involved in WNV transmission as well as environmental conditions can vary across sites, we decided not to pursue this route and focus the present study on the current gaps. It follows that some very high values we found for $R_0$ and $R_t$ can be considered as an overestimate caused by the extreme (and possibly not biologically plausible) values of parameters. Therefore, this work should not be considered as a framework to predict how WNV spreads in Lombardy region, but as an analysis of both where the current major gaps are, and to what extent these gaps affect our ability to predict WNV spread.

In conclusion, WNV transmission and maintenance processes show knowledge gaps, thus impairing our ability to understand and predict its spread. Among them, the duration of the avian infectious period and the fraction of bites on competent birds are the most impactful factors driving WNV circulation. Therefore, they can be considered among the most important parameters to be further studied, and their investigation could also help in determining the avian species that play the main role in WNV spread and maintenance in northern Italy. Furthermore, temperature and mosquito abundance can be limiting factors for WNV spread, and areas with suitable conditions require the design of an efficient surveillance plan to keep disease spread under control. Finally, our results, obtained through mathematical model simulations, highlight how a synergic interaction among theoretical and field research could be beneficial for a better understanding of infectious disease transmission processes by allowing

the formulation of hypotheses to identify the most appropriate data required to cover knowledge gaps.

## Supporting information

**S1 Appendix. Supplementary information.**
(PDF)

**S1 Table. Recorded average entomological captures for each year and cluster.**
(XLSX)

**S2 Table. Total number of analysed mosquito pools for each year and cluster.**
(XLSX)

**S3 Table. Total number of WNV-positive pools for each year and cluster.**
(XLSX)

## Author Contributions

**Conceptualization:** Elisa Fesce, Giovanni Marini, Mario Chiari, Marco Farioli, Nicola Ferrari.

**Data curation:** Elisa Fesce, Giovanni Marini, Monica Pierangela Cerioli.

**Formal analysis:** Elisa Fesce.

**Investigation:** Davide Lelli, Mario Chiari, Marco Farioli.

**Methodology:** Giovanni Marini.

**Project administration:** Nicola Ferrari.

**Software:** Elisa Fesce, Giovanni Marini.

**Supervision:** Roberto Rosà, Nicola Ferrari.

**Validation:** Giovanni Marini.

**Visualization:** Elisa Fesce, Giovanni Marini, Nicola Ferrari.

**Writing – original draft:** Elisa Fesce.

**Writing – review & editing:** Giovanni Marini, Roberto Rosà, Davide Lelli, Monica Pierangela Cerioli, Mario Chiari, Marco Farioli, Nicola Ferrari.

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
