## [Decision Letter · Decision Letter 0]

14 Jun 2022

Dear Mrs Fesce,

Thank you very much for submitting your manuscript "Understanding West Nile virus spread: Mathematical modelling to determine the mechanisms with the most influence on infection dynamics." for consideration at PLOS Neglected Tropical Diseases. As with all papers reviewed by the journal, your manuscript was reviewed by members of the editorial board and by several independent reviewers. In light of the reviews (below this email), we would like to invite the resubmission of a significantly-revised version that takes into account the reviewers' comments. 

The Authors are expected to address all the criticisms by all Reviewers. In particular, please provide more detailed description of the MCMC procedures, including convergence, autocorrelation analysis and thinning (Reviewers #1 & #2), consider the potential dependence between the 4 parameters of investigation (Reviewer #1), provide further justification for the selection of the 4 key parameters for assessment, reconsider the connection of the range for parameter assessment to the realistic parameter range (Reviewer #2) and clarify how Rt was derived and connected to time-dependent parameters (Reviewer #3). In additional to the above comments, please address,

1. What are the joint impact of the 4 key parameters? could some of the parameters may have synergistic effect on transmission? Some assessment (at least pairwise) would provide more insights from the study.

We cannot make any decision about publication until we have seen the revised manuscript and your response to the reviewers' comments. Your revised manuscript is also likely to be sent to reviewers for further evaluation.

Sincerely,

Eric HY Lau, Ph.D.

Associate Editor

Benjamin Althouse

Deputy Editor

The Authors are expected to address all the criticisms by all Reviewers. In particular, please provide more detailed description of the MCMC procedures, including convergence, autocorrelation analysis and thinning (Reviewers #1 & #2), consider the potential dependence between the 4 parameters of investigation (Reviewer #1), provide further justification for the selection of the 4 key parameters for assessment, reconsider the connection of the range for parameter assessment to the realistic parameter range (Reviewer #2) and clarify how Rt was derived and connected to time-dependent parameters (Reviewer #3). In additional to the above comments, please address,

1. What are the joint impact of the 4 key parameters? could some of the parameters may have synergistic effect on transmission? Some assessment (at least pairwise) would provide more insights from the study.

Reviewer's Responses to Questions

**Key Review Criteria Required for Acceptance?**

**Methods**

-Are the objectives of the study clearly articulated with a clear testable hypothesis stated?

-Is the study design appropriate to address the stated objectives?

-Is the population clearly described and appropriate for the hypothesis being tested?

-Is the sample size sufficient to ensure adequate power to address the hypothesis being tested?

-Were correct statistical analysis used to support conclusions?

-Are there concerns about ethical or regulatory requirements being met?

Reviewer #1: (No Response)

Reviewer #2: (No Response)

Reviewer #3: 2 - The authors should clearly stated throughout the paper that this study just deal with single mosquito species and, most importantly, single bird species. 

3 - R_t estimate: the rational and derivation are not clear and need to be explained. Some included parameters are supposed to change over time and it is hardly understandable that the final expression of R_r looks very similar to R_0

4 - Provide a clear definition of the spread probability and how is this computed. In addition, I would like to remind to the authors that the probability of infection spread is not "1" even for R_0 >1.

**Results**

-Does the analysis presented match the analysis plan?

-Are the results clearly and completely presented?

-Are the figures (Tables, Images) of sufficient quality for clarity?

Reviewer #1: (No Response)

Reviewer #2: (No Response)

Reviewer #3: 5 - Figues in the Appendix are missing

6 - S1 Appendix, Table A: Something missing in the expression or value of Theta_M

7 - Figure quality should be improved

**Conclusions**

-Are the conclusions supported by the data presented?

-Are the limitations of analysis clearly described?

-Do the authors discuss how these data can be helpful to advance our understanding of the topic under study?

-Is public health relevance addressed?

Reviewer #1: (No Response)

Reviewer #2: (No Response)

Reviewer #3: (No Response)

**Editorial and Data Presentation Modifications?**

Reviewer #1: (No Response)

Reviewer #2: (No Response)

Reviewer #3: (No Response)

**Summary and General Comments**

Reviewer #1: In this manuscript titled “Understanding West Nile virus spread: Mathematical modelling to determine the mechanisms with the most influence on infection dynamics”, the authors investigate the mechanisms most affecting the transmission of Culex pipiens-borne West Nile disease by analysing a mathematical model of vector and transmission dynamics. The authors modify the model published in Marini et al. (2020), incorporate recently collected field data, and perform sensitivity analysis on a subset of parameters relevant to disease transmission.

Several vector species, their respective biting preferences, and reservoir species are involved in sustaining and spreading the virus. Several approaches for modelling the underlying mechanisms have been proposed (by the same group and others such as Bergsman et al. 2016). The authors present a technical analysis of a particular model, but not sufficiently novel evidence to improve the parameterisation of the model or the understanding of the disease mechanism. One might wonder if the authors would obtain the same conclusions, had they used a different model. Overall, in its current state, the manuscript seems appropriate for a more technical audience.

The authors adapted a recently published model to perform the analysis, giving more emphasis to the mosquito-bird interactions. However, it is not clear in the text why this adaptation is necessary, and why the analysis could not be performed on the original model. It is also not clear how the model was adapted. An extended description of the changes would add significantly to the clarity of the manuscript.

A major issue in the analysis is the authors’ statement of performing 10,000 iterations of MCMC to sample 10,000 parameter sets from the posterior distribution, which implies that the initial transient state of the MCMC chain was not discarded, autocorrelation was not checked, and thinning (where every n-th iteration is selected to prevent autocorrelation) was not applied. Although the authors state that a random sub-sample of 100 parameters sets were drawn from this reservoir set, neither these sub-samples nor the original sample can be regarded as proper samples from the posterior distribution without the validation steps mentioned above.

The authors performed the sensitivity analysis on 4 parameters. Although the choice of these are justified, due to the uncertainty in the other parameters (due to experimental errors or uncertainties in inference), a broader analysis including other relevant parameters might also be helpful. More importantly, the analysis was performed on each parameter independently, however, their potential interdependencies necessitate investigating the impact of pairwise or random perturbations on all parameters.

Minor points:

- The statement in line 440 should not be interpreted in the biological context. A few initial infectious mosquitoes and a low number of birds might be enough to start and maintain the infection at the beginning of the season in the context of this deterministic model, but hardly ever any biological conclusions can be drawn from this. When geospatial range and stochastic interactions dominate, the probability of sustaining the infection may decrease considerably.

- The Supplementary Figure S1 detailing the goodness of fit could have been more informative had it resolved the values less than 5, where most of the data seems to be.

- Having a different order of the same parameters in Figures 4 and 5 is confusing.

- I noticed a typo in S1 Appendix Table A, the mosquito death rate parameter 4.75 was originally given as 4.57 in Marini et al. (2020).

Reviewer #2: The authors present a sensitivity analysis of an existing mathematical model to identify the parameters which are the most influential in determining the risk of West Nile virus transmission. The study also estimates these parameters based on field data collected in Northern Italy. The authors are certainly correct that the parameters they study are poorly understood, particularly in a European context, and so highlighting the importance of these parameters given the results of the mathematical model is an important and worthwhile endeavour. It must be noted that many of the results highlighted may already be known to those familiar with R0 (or time-varying R0) equations for vector-borne diseases and their interpretation (for example, biting rate is relatively important because it appears as a squared term in the R0 equation where the terms governing transmission probabilities are not squared); however, as this may be a relatively small subset of the readership of the journal the paper is still likely to be of reasonably board interest and appeal. Further, the estimates of these parameters based on the field data (whilst very variable in some cases) will be of wide interest and may benefit future modelling studies. Nonetheless, I do have some concerns about some of the analyses and presentation which I feel need addressed prior to publication. Full details of my concerns and suggestions are given below:

- I would suggest that a more thorough justification for the 4 parameters chosen for further study is needed, as this choice is clearly central to the paper and very important. At present there is little justification beyond the authors stating they are the most important. Whilst I personally mostly agree, I still think more explanation should be given (why not host to host transmission or vertical transmission, for example?). In particular, I would question why the duration of the incubation period of the virus within the mosquito was not included as, to my knowledge, there is substantial uncertainty surrounding this parameter?

- The authors choose the parameters p (avian competence to infection) and p_MB (avian susceptibility to infection) for inclusion in their analysis and the analyses then centre around the calculation of Rt (lines 246-249) and understanding how percentage changes in the parameters will affect Rt. It can be seen from the equation that both of these parameters (p and p_MB) will have a linear effect on Rt (e.g. a 50% decrease in either parameter will cause a 50% decrease in Rt). This is indeed what the authors go on to show in Figure 4 but it begs the question of why to include both parameters when they are essentially identical within the model (albeit with a small difference when calculating spread probability due to the thresholding of Rt at 1)?

- Related to the above point, both of the parameters p and p_MB are defined as probabilities with baseline values of 0.62 and 0.78 respectively (lines 204-207 and Table 1), so a 90% increase in these parameters would lead to a probability greater than 1. I had initially assumed that the values would be capped to 1, however the sustained linear increase in Figure 4 suggests not. Am I misunderstanding something or are there probabilities >1?

- I worry that there is a bit of a problem with the approach of considering percentage changes for each of the parameters. A lot of what comes out as important here depends on the model structure. For example, the authors have identified the bird recovery rate as very important, however much of this stems from the fact that it appears in the denominator of the Rt equation and so as the recovery rate approaches 0 Rt will approach infinity. Thus, if we consider a 90% reduction from the baseline then the recovery rate gets close to zero (it takes nearly 4 weeks to recover) and we see a much bigger increase in Rt than for the other parameters which appear in the numerator; however, if we consider a 90% increase from the baseline estimates then recovery rate actually results in the smallest difference of the 4 parameters but this difference is less noticeable when plotted. This 90% decrease in the recovery rate results in a value beyond the lower end of the 95% interval stemming from the MCMC, which begs the question of whether or not this is really the most important parameter if the biggest effect on Rt comes when we reach values which appear highly unlikely given the field data? Why not investigate changes at fixed centiles across the range of the 95% interval from the MCMC? This way the changes for each parameter are across the range of the most plausible values for each parameter? The results might be similar but I feel there is some nuance being missed here whereby, for example, the infectious period in birds may only be particularly influential if the true value is in fact very low (and possibly lower than the field data would suggest).

- I see that the authors have addressed some of the concerns in my previous point in the discussion where they highlight that the bird recovery rate is only particularly influential if it is in fact very low. This discussion was very welcome and I feel that this caveat should be made more apparent elsewhere. For example, at the start and end of the discussion the authors state that the bird recovery rate is very important without stating this caveat and I feel it should be included in those places.

More minor comments and suggestions:

- Line 67 – “approximately 25%”

- Lines 70-71 – say when WNV was introduced for the first time?

- Line 76 – delete encountered

- Line 82 – “potentially inaccurate” since we don’t actually know how inaccurate (if at all) they will be

- Lines 90-93 – maybe add a reference?

- Daily mean temperatures have been used throughout. Given what we know about the importance of diurnal temperature variation I wondered why that was not incorporated?

- I think there should be some more details about the MCMC procedure given. For example, say something about checking for convergence and/or autocorrelation in the chains? I'd also like to know how the priors were chosen - I couldn't find this here or in a brief scan of the previous paper.

- The assumption that birds do not lose immunity seems likely to be a big and potentially influential one. It would be good if there was a reference to support birds maintaining immunity for a long time. If there is no such reference because it has not been studied then would it not be important to consider a further scenario where birds lost their immunity?

- Lines 231 and 249 – what is delta_B here? I can’t find it defined. I’m assuming this is a typo and it should be the bird recovery rate, nu_B.

- Lines 299-302 - Suggest rewording to something like "No meaningful correlation" considering you filter on both p-value and correlation strength. If, for example, r=0.3 and p=0.01 then there is reasonable evidence there might be a correlation, just one that is weak enough that it may not be interesting. Also, why are the last two of the three correlations mentioned included? They have tiny correlation values and high p-values and don't meet the criteria for a correlation specified above.

- Lines 309 – 320 and Fig 4 – The figure labelling has gone wrong here. The (A)-(D) in the text don’t match up with the Figure.

- Line 328 – “the least” rather than “a less”

- Line 330 – The authors highlight August/September here but I think May looks similar.

- Line 331 – I think August and May look just as variable as July and September.

- Line 333 – “showed similar results” – Given the two parameters have the same effect on Rt then I think these results will be identical, no?

- Lines 333-334 - I think the authors mean increased here because the first half of the sentence talks about a decrease. If we are considering the increase then the effect in p will be slightly larger than p_MB because the baseline value of p is slightly lower and as these are probabilities there is a threshold on how much they can be increased. So I think this may be the only case where I would expect that there may indeed be a small difference between the two parameters (unless I’m fundamentally misunderstanding something). I would still like some confirmation that these probabilities are not being allowed to become greater than 1 though.

- Line 343 – I find this sentence confusing. I’m not sure what the authors are referring to when they say “lower than that for”. What is “that” in this context?

- Lines 439-441 – I’m struggling to remember where this was shown. Include a reference to the relevant Figure or Table.

- There's very little known about these parameters in Europe but there is a bit more in North America. It might be interesting to add a paragraph in the discussion comparing the values here to some of the values there? This is very much just a suggestion though, if the author's disagree then I am happy to be ignored.

Reviewer #3: This paper deals with the understanding of the spread of West Nile Virus (WNV) in Lombardi, northern Italy. The authors propose a sensitivity analysis to identify the parameters with the largest impact on the spread of the WNV infection. They used a mathematical model simulating WNV spread to estimate the basic reproduction number of the infection and to quantify the infection spread in mosquitoes (Culux pipes only) and birds (magpies). The focus was on how the following four parameters might affect WNV transmission: i) mosquito biting rate on birds, ii) avian competence, iii) the duration of the infectious period in birds or avian recovery and iv) avian susceptibility to WNV infection. They found that the duration of the infectious period in birds and mosquito biting rate are the most impactful parameters. 

Although all of this is not new at all, the paper is well written and report interesting results that are worthwhile to be published. Therefore, I would recommend to publish this material in PNTD after the authors have considered the following revisions.

1 - Title: There is no new mechanism of WNV transmission. The authors rather carried out an estimation of some parameters of the problem. Revise the title to reflect that.

2 - The authors should clearly stated throughout the paper that this study just deal with single mosquito species and, most importantly, single bird species. 

3 - R_t estimate: the rational and derivation are not clear and need to be explained. Some included parameters are supposed to change over time and it is hardly understandable that the final expression of R_r looks very similar to R_0

4 - Provide a clear definition of the spread probability and how is this computed. In addition, I would like to remind to the authors that the probability of infection spread is not "1" even for R_0 >1. 

5 - Figues in the Appendix are missing

6 - S1 Appendix, Table A: Something missing in the expression or value of Theta_M

7 - Figure quality should be improved

PLOS authors have the option to publish the peer review history of their article (what does this mean?). If published, this will include your full peer review and any attached files.

Reviewer #1: No

Reviewer #2: Yes: David Ewing

Reviewer #3: No
---

## [Decision Letter · Decision Letter 1]

1 Dec 2022

Dear Mrs Fesce,

Thank you very much for submitting your manuscript "Understanding West Nile virus transmission: mathematical modelling to quantify the most critical parameters to predict infection dynamics" for consideration at PLOS Neglected Tropical Diseases. As with all papers reviewed by the journal, your manuscript was reviewed by members of the editorial board and by several independent reviewers. In light of the reviews (below this email), we would like to invite the resubmission of a significantly-revised version that takes into account the reviewers' comments. 

The Authors are expected to address all the criticisms by all Reviewers. In particular, please confirm the interpretation of the parameters, and the associations between parameters have to be considered in the analyses and interpretation (Reviewer #1). In additional to the above comments, please address,

1. Abstract, “This analysis allows us to identify the investigated parameters that need the most accurate estimate and to further investigate transmission processes that majorly affect WNV spread.” In Table 1, the estimated recovery rate had a very wide 95% CI (0.002-0.94). Most of the parameters had very wide 95% CI too almost covering >80% of the plausible parameter range. Please take this into account and revise the abstract/discussion.

2. Table 1, all the credible intervals for the estimated parameters covered almost the entire range of the prior distributions. This is probably an indication that the model and data could not provide reliable estimates for any of the parameters. One possible improvement is to use more information priors based on the literature. This is important as quantification of parameters is a main study objective but this is not yet achieved.

3. Table 2 shows the upper bound of some parameters would lead to implausible R0 (>1000). This is probably an issue related to the associations between parameters and hence parameters should not be analyzed independently.

We cannot make any decision about publication until we have seen the revised manuscript and your response to the reviewers' comments. Your revised manuscript is also likely to be sent to reviewers for further evaluation.

Sincerely,

Eric HY Lau, Ph.D.

Academic Editor

Benjamin Althouse

Section Editor

The Authors are expected to address all the criticisms by all Reviewers. In particular, please confirm the interpretation of the parameters, and the associations between parameters have to be considered in the analyses and interpretation (Reviewer #1). In additional to the above comments, please address,

1. Abstract, “This analysis allows us to identify the investigated parameters that need the most accurate estimate and to further investigate transmission processes that majorly affect WNV spread.” In Table 1, the estimated recovery rate had a very wide 95% CI (0.002-0.94). Most of the parameters had very wide 95% CI too almost covering >80% of the plausible parameter range. Please take this into account and revise the abstract/discussion.

2. Table 1, all the credible intervals for the estimated parameters covered almost the entire range of the prior distributions. This is probably an indication that the model and data could not provide reliable estimates for any of the parameters. One possible improvement is to use more information priors based on the literature. This is important as quantification of parameters is a main study objective but this is not yet achieved.

3. Table 2 shows the upper bound of some parameters would lead to implausible R0 (>1000). This is probably an issue related to the associations between parameters and hence parameters should not be analyzed independently.

Reviewer's Responses to Questions

**Key Review Criteria Required for Acceptance?**

**Methods**

-Are the objectives of the study clearly articulated with a clear testable hypothesis stated?

-Is the study design appropriate to address the stated objectives?

-Is the population clearly described and appropriate for the hypothesis being tested?

-Is the sample size sufficient to ensure adequate power to address the hypothesis being tested?

-Were correct statistical analysis used to support conclusions?

-Are there concerns about ethical or regulatory requirements being met?

Reviewer #1: (No Response)

Reviewer #2: (No Response)

Reviewer #3: (No Response)

**Results**

-Does the analysis presented match the analysis plan?

-Are the results clearly and completely presented?

-Are the figures (Tables, Images) of sufficient quality for clarity?

Reviewer #1: (No Response)

Reviewer #2: (No Response)

Reviewer #3: (No Response)

**Conclusions**

-Are the conclusions supported by the data presented?

-Are the limitations of analysis clearly described?

-Do the authors discuss how these data can be helpful to advance our understanding of the topic under study?

-Is public health relevance addressed?

Reviewer #1: (No Response)

Reviewer #2: (No Response)

Reviewer #3: (No Response)

**Editorial and Data Presentation Modifications?**

Reviewer #1: (No Response)

Reviewer #2: (No Response)

Reviewer #3: (No Response)

**Summary and General Comments**

Reviewer #1: I thank the authors for their efforts in preparing this improved version of the manuscript. I am pleased to notice that most of my concerns have been adequately addressed. However, there still remains a couple of major issues to be addressed in further revision.

The authors changed the statement "Moreover, our model shows that few initial infectious mosquitoes and a low number of birds are enough to start and maintain the infection at the beginning of the season" into "Moreover, we showed that a low vector-host ratio (100 mosquitoes/host) and an initial number of competent hosts lower 5 (table 1) are enough to start and maintain the infection at the beginning of the season." and added "Also, the model provided is deterministic, thus not taking into account any stochasticity of events." later in the paragraph. It is not clear to me how the authors derive this conclusion about the impact of vector-host ratio and initial number of competent hosts on R0 at the beginning of the season. Nevertheless, these changes do not address my earlier concerns that a deterministic model is being interpreted erroneously.

Related to this issue, in line 289, 357, and in the Results section "Transmission maintenance during seasons", the frequency of simulations (obtained from the posterior distribution of parameter estimates) leading to R0 > 1 is used to indicate the frequency (or probability) of an outbreak. I believe the "outbreak frequency" referred by the authors is in fact an estimation of R0, and the range of R0 simulated using the posterior samples is the range in R0. In this case, interpreting the quantity as outbreak frequency (or probability) is misleading. I suggest that authors display the value of R0 and its variability, and assess risk based on the times when the average R0 is more than 1. This also applies to the interpretation of Rt in the final section of Results.

The authors claim that there are no associations among the transmission parameters. However, they have chosen to investigate a subset among all the inferred parameters. Evidently, these have no significant associations among them. The posterior samples clearly show, as the authors have also displayed in S1 Fig. F, there are significant associations among the inferred transmission parameters including the following: iBw and iBe are strongly positively correlated, and ai and nuB are strongly negatively correlated. Perhaps the authors mean "no association between the bird-related transmission parameters chosen for analysis in this context".

Minor points

=========

In line 183, the reference cited is a book by Anderson titled "Biology of the Ubiquitous House Sparrow: From Genes to Populations". I am not able to confirm any reference to mosquito biting rate quantification in this book. However, I urge the authors to double-check if this is the intended citation.

Also, in lines 203-4, the proposed likelihood function does not appear in the cited manuscript #25 (Moschini et al. 2017).

In line 238, all prior distributions pertaining the epidemiological model are reported, but I couldn't see any information on K1, K2, and M0. The authors should state their interest in investigating the epidemiological model at this point.

In line 299, the authors chose to refer the amplitude of parameters perturbations as CI, the confidence interval. I strongly recommend that the authors choose a different term for this range, as CI has already been widely misused in the literature.

In line 353, the grey boxplot visualising the posterior distribution seems to be missing in Figure 4.

In the first paragraph of Discussion, the authors claim that they proved the importance of having a reliable and precise estimate of the bird recovery rate, but the presentation of this argument could be much improved. I believe this is a strong statement, and should be paraphrased. 

In line 487, it is not clear how the temperatures and mosquito densities in Lombardy always sustain WNV spread. Does this apply to every year, every time of the year, and even accounting for diurnal variability? I don't think such a strong statement is supported by the evidence presented.

In S1 Fig, H, the parameters analysed (f and pVB) do not match with the earlier description of model parameters. What is f in relation to f1 and f2? Is pVB the same as pMB?

The parameter "b" is not included in model description. Also the description of "p" is left out of Appendix S1. I believe the complete description of the model and equations, as claimed by the authors in lines 195-196, should include all such details and variations in parameters.

My final comment is that the manuscript seems to be written by multiple authors with different styles/experiences, and this seems to have disrupted coherence in certain parts. Other than this, I enjoyed reading this interesting manuscript.

Reviewer #2: I would like to thank the authors for their consideration of my comments and their work in editing the manuscript. I am now happy to recommend the work for publication. My only remaining comment is that I have selected "No" in the data availability drop-down menu because I don't think the model code is publically available, which in the name of open science I think it should be.

Reviewer #3: (No Response)

PLOS authors have the option to publish the peer review history of their article (what does this mean?). If published, this will include your full peer review and any attached files.

Reviewer #1: No

Reviewer #2: Yes: David Ewing

Reviewer #3: No
---

## [Decision Letter · Decision Letter 2]

1 Apr 2023

Dear Fesce,

We are pleased to inform you that your manuscript 'Understanding West Nile virus transmission: mathematical modelling to quantify the most critical parameters to predict infection dynamics' has been provisionally accepted for publication in PLOS Neglected Tropical Diseases.

Best regards,

Eric HY Lau, Ph.D.

Academic Editor

Benjamin Althouse

Section Editor

Reviewer's Responses to Questions

**Key Review Criteria Required for Acceptance?**

**Methods**

-Are the objectives of the study clearly articulated with a clear testable hypothesis stated?

-Is the study design appropriate to address the stated objectives?

-Is the population clearly described and appropriate for the hypothesis being tested?

-Is the sample size sufficient to ensure adequate power to address the hypothesis being tested?

-Were correct statistical analysis used to support conclusions?

-Are there concerns about ethical or regulatory requirements being met?

Reviewer #1: (No Response)

**Results**

-Does the analysis presented match the analysis plan?

-Are the results clearly and completely presented?

-Are the figures (Tables, Images) of sufficient quality for clarity?

Reviewer #1: (No Response)

**Conclusions**

-Are the conclusions supported by the data presented?

-Are the limitations of analysis clearly described?

-Do the authors discuss how these data can be helpful to advance our understanding of the topic under study?

-Is public health relevance addressed?

Reviewer #1: (No Response)

**Editorial and Data Presentation Modifications?**

Reviewer #1: (No Response)

**Summary and General Comments**

Reviewer #1: I thank the authors for this improved version. I confirm that most of my concerns has been addressed, and I am pleased to recommend the manuscript for publication as is.

PLOS authors have the option to publish the peer review history of their article (what does this mean?). If published, this will include your full peer review and any attached files.

Reviewer #1: No

---

## [Editor Report · Acceptance letter]

26 Apr 2023

Dear Fesce,

We are delighted to inform you that your manuscript, "Understanding West Nile virus transmission: mathematical modelling to quantify the most critical parameters to predict infection dynamics," has been formally accepted for publication in PLOS Neglected Tropical Diseases.

Best regards,

Shaden Kamhawi

co-Editor-in-Chief

Paul Brindley

co-Editor-in-Chief
